

# Status of coral reefs in Antigua & Barbuda: using data to inform management

Ruleo Camacho[1], Sophia Steele[2], Shanna Challenger[3] and Mark Archibald[4]

[1] Department of Environment, Government of Antigua and Barbuda, St. John, Antigua and Barbuda
[2] Fauna & Flora International, Cambridge, UK
[3] Redonda Restoration Program, Environmental Awareness Group, St. John, Antigua and Barbuda
[4] Fisheries Division, Government of Antigua and Barbuda, St. John, Antigua and Barbuda

## ABSTRACT

The nation of Antigua and Barbuda has experienced major degradation of its coral reef ecosystems over the past 40+ years. The primary drivers of this degradation are multiple and are highly linked to anthropogenic influences, including over-exploitation and poor management of marine resources. The effectiveness of management actions in marine protected areas (MPAs) has often been hampered by a lack of data to inform management recommendations. This was emphasized by The Nature Conservancy's (TNC) Coral Reef Report Card which highlighted not only the lack of data collection in Antigua and Barbuda and other Caribbean nations, but also illustrated how spatially dispersed available datasets are. The government of Antigua and Barbuda recognized the need for a marine data collection program to better inform the designation and management of MPAs as a tool to improve the health of the marine ecosystems. The Atlantic Gulf Rapid Reef Assessment (AGRRA) protocol has been identified as a means to address planning and management for marine areas. Three AGRRA surveys have been conducted in the years following the TNC 2016 report, in previously established managed areas: North East Marine Management Area (NEMMA) in 2017 and Nelson Dockyard National Park (NDNP) in 2019 as well as areas outlined for future management (Redonda in 2018). Our surveys were conducted to provide updated datasets to inform management for the aforementioned areas. While the results of these surveys mirror the underlying poor coral reef-health conditions, which have been shown to exist within the Caribbean region, they also highlight intra-site variation that exists within each survey location. This knowledge can be crucial in guiding management decisions in these marine areas, through zoning and other management prescriptions. Additionally, the marine surveys conducted around Redonda established useful marine baselines to aid in monitoring the island's recovery following removal of terrestrial invasive species. This article provides an overview of data collected using the AGRRA methodology in marine zones across Antigua and Barbuda which have current or future management prescriptions and provides recommendations to demonstrate the data's future utilization for marine conservation and management.

Corresponding author
Ruleo Camacho,
rcam.doe@gmail.com

# INTRODUCTION

Coral reef ecosystems in the Caribbean have been subject to a phase-shift from coral-dominated to algal-dominated ecosystems (*Hughes, 1994*; *Mumby, Hastings & Edwards, 2007*; *Mumby & Steneck, 2008*; *Mumby et al., 2012*; *Jackson et al., 2014*; *Steneck et al., 2018*) over the past 40 years, a shift that has been reflected in the reefs of Antigua and Barbuda (*Camacho & Steneck, 2016*; *Kramer et al., 2016*). Marine Protected Areas (MPAs), are one of the tools used to counter the decline of coral reef ecosystems around the world (*Guarderas, Hacker & Lubchenco, 2008*; *Bustamante et al., 2014*) by implementing regulations to reduce anthropogenic stress. However, the lack of both data-driven goals and an effective management structure can often result in MPAs which fail to meet the objectives for which they were set up (*McClanahan, 1999*; *Kaplan et al., 2015*; *Camacho & Steneck, 2016*). The Nature Conservancy (TNC), in 2016, combined existing datasets available in the literature for the Caribbean region and published coral reef report cards for six Caribbean countries (St. Kitts and Nevis, Antigua and Barbuda, Dominica, St. Lucia, St. Vincent and the Grenadines and Grenada) (*Kramer et al., 2016*). These report cards provide an overview of the coral reef health parameters, while identifying gaps in the data available to decision-makers within these Small Island Developing States (SIDS) participating countries. To rate the health of coral reefs throughout the Caribbean, TNC used a Reef Health Index (RHI) (Table 1). The RHI scale uses four parameters (Coral Cover, Fleshy Macroalgae, Commercial Fish Biomass, Herbivorous Fish) to enhance reef managers understanding of the conditions affecting their reef systems, recommend management prescriptions, and provide a useful comparison ranking. Within the RHI, Antigua and Barbuda ranked "poor" overall, particularly as it related to coral cover, fleshy macroalgae and commercial fish biomass, while herbivorous fish biomass ranked "fair" (Table 2). Additionally, the report card highlighted the lack of regularity (last data collection in 2013) and evenness or spread of data collection on coral reefs in Antigua and Barbuda. With 22 designated managed marine areas on the books (*GoAB, 2019*), and additional areas proposed, there is an apparent need to have updated ecological information to guide the management of these marine resources. The Government of Antigua and Barbuda (GoAB) recognized that implementing a data monitoring program could identify marine ecological issues, inform decision-making and MPA management planning, and assist with reporting requirements for Multilateral Environmental Agreements (MEAs) such as the Convention on Biological Diversity (CBD). Due to its longstanding regional network, the Atlantic Gulf Rapid Reef Assessment (AGRRA) methodology (*Lang et al., 2017*) was identified as the primary method of coral reef data collection for the island. In addition, the availability of trainers within the region and the rapid analysis of datasets and comparability with

**Table 1 Reef Health Index (RHI) values.**

| The reef health index (RHI) | Reef health index reference values | | | | |
|---|---|---|---|---|---|
| | Critical 1–1.8 | Poor 1.9–2.6 | Fair 2.7–3.4 | Good 3.5–4.2 | Very good 4.3–5 |
| Coral cover (%) | <5 | 5.0–9.9 | 10.0–19.9 | 20.0–39.9 | >=40 |
| Fleshy macroalgal cover (%) | >25.0 | 12.1–25 | 5.1–12.0 | 1.0–5.0 | 0–0.9 |
| Herbivorous fish (g/100 m$^2$) | <960 | 960–1,919 | 1,920–2,879 | 2,880–3,479 | >=3,480 |
| Commercial fish (g/100 m$^2$) | <420 | 420–839 | 840–1,259 | 1,260–1,679 | >=1,680 |

**Table 2 Reef Health Index (RHI) comparison.**

| Indicator | Year | Score | Average | Year | Score | Average | Trend |
|---|---|---|---|---|---|---|---|
| Coral cover (%) | 2015 | Poor | 9 | 2019 | Poor | 9 | No change |
| Fleshy macroalgae (%) | 2015 | Poor | 18 | 2019 | Poor | 23 | Negative |
| Herbivorous fish (g/100 m$^2$) | 2015 | Fair | 2,810 | 2019 | Fair | 2,765 | Negative |
| Commercial fish (g/100 m$^2$) | 2015 | Poor | 500 | 2019 | Good | 1,914 | Positive |

Note:
Values for 2016 were presented from the summarization of previously existing data by TNC. Values for 2019 are the summarization of AGRRA values collected in the years 2017, 2018 and 2019 that are reported in this article.

previous data collections both locally and regionally, made it an ideal method to establish baseline coral reef data.

Three AGRRA assessments, conducted between 2017 and 2019, are reported in this article (Fig. 1A). These surveys were collected at the request of various projects/departments and the availability of relevant funding resources. They all possess the common theme of being located within areas which are currently managed or have been identified as an area of future management. These surveys were not chosen to fill in all data gaps across the islands' entire geographic area, but as a means of enriching the database for marine ecological conditions around Antigua and Barbuda as part of an on-going effort of baseline data collection and coral reef monitoring. Our findings highlight variation both among assessments conducted at different parts of the island, as well as within assessments. Understanding these site-specific differences is crucial to enhancing our knowledge of their associated marine ecology and will provide insight into the most appropriate management prescriptions for each area.

# MATERIALS AND METHODS

## Site descriptions

North-East Marine Management Area (NEMMA): This site was declared as a Marine Protected Area in 2005 under the Fisheries Act (1983) and the amended Fisheries Act (2006) (*Jackson, 2008*) and has a marine area of 108.5 km$^2$, making it the largest within the waters of Antigua and Barbuda. Its long-outdated management plan (*Jackson, 2008*) requires review and renewal (T. Lovel & Fisheries Division, 2018, personal communications), and currently has no location-specific enforcement actions other than

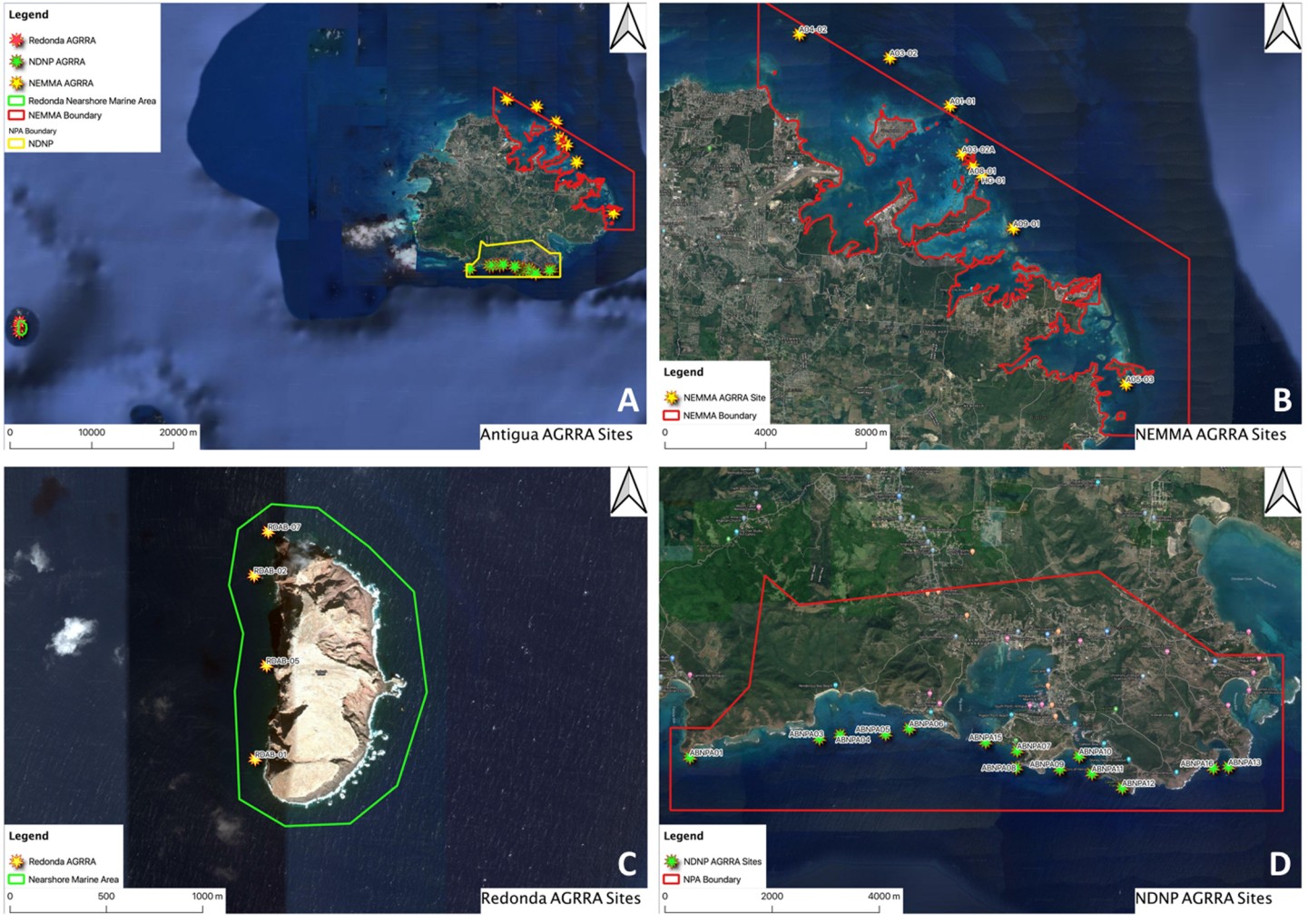

**Figure 1 Map of study sites.** AGRRA sites surveyed: (A) All Antigua and Barbuda sites between 2017 and 2019; (B) NEMMA sites; (C) Redonda Sites; (D) NDNP Sites. Maps created by Ruleo Camacho using QGIS software.

general fishery regulations. According to the TNC 2016 Coral Reef Report Card, NEMMA is located within sub-region 33, which is described as "indenting coastline with a wide shelf and greatest coral reef development" (*Kramer et al., 2016*). To the east, the NEMMA faces the full force of the Atlantic Ocean (Fig. 1B), while to the West, the coastline is a combination of mangrove wetlands, rocky shorelines and over 30 small offshore islands. On the terrestrial side of NEMMA, there are several industrial (inclusive of Antigua Power Company, Parham Fisheries Complex, Shell Beach Marina and Jumby Bay Resort), recreational (Stingray City and Antigua Nature Tours) and residential areas. While the protected portions only encompass the marine and coastal areas, the adjacent developments undoubtedly have direct and indirect effects on the area. The protected area has a combination of patch and fringing reefs, with the inner areas dominated by seagrass beds and sandy flats. The data presented in this article were collected as an update to the benthic ecological conditions in the area (*Palmer, 2017*) and were based on surveys conducted in 2005 by a team from the University of Miami (*Brandt et al., 2005*).

Redonda: The island of Redonda is located 48 km southwest of the mainland Antigua. Although geographically closer to the islands of St. Kitts (28 km) and Montserrat (19 km), it is politically recognized as a territory of Antigua and Barbuda. The island has been uninhabited since the 19th century, when it was used for guano mining due to the high seabird population and is recognized as an Important Bird and Biodiversity Area (IBA—AG001) for its significant populations of nesting Boobies (Sulidae family). The island is surrounded by cliffs, with no safe coastal access. The marine landscape reflects a similar situation, with depths of less than 15 m limited to a maximum distance of 150 m from the island but averaging within 100 m from the shore (defined as nearshore marine area hereafter) (Fig. 1C). The nearshore marine areas are dominated by boulder reefs, except for a western portion which is home to "spur and groove" reef formations. Outside of these reef areas are seagrass beds sloping into deeper habitat. The island of Redonda has undergone tremendous terrestrial interventions (Redonda Restoration Program—RRP) (*Bell et al., 2017*) to remove invasive alien species (IAS) (rats and goats) and has so far resulted in remarkable recovery of the terrestrial fauna and flora (S. Challenger, 2019, personal observations). Redonda and its surrounding seas are currently under review for legal declaration as a Protected Area under the Environmental Protection and Management Act (2019) legislation. The total proposed area of this management zone is 299 km$^2$ with an average depth of ~60 m, but the area available for survey using the AGRRA methodology is ~2 km$^2$ due to depth and safe diving limitations. There is no current human settlement on Redonda, or any plan for this in the future. Access to the terrestrial landscape is restricted to helicopter access due to its sheer cliffs and unstable terrain. Baseline marine data were required to guide the development of the management plan for Redonda and its surrounding waters. The data will also aid in the study of impacts of the terrestrial recovery on the associated marine ecosystem as similar activities in other countries have demonstrated increases in reef productivity (*Graham et al., 2018*). Due to its small size and location away from the mainland, marine work around the island is difficult, as its total exposure, "wrap-around" currents and the lack of safe anchorage makes it a relatively unsafe environment for work except in the best of weather conditions. Surveys in 2018 were limited to the relatively protected side of the island due to adverse weather. Redonda has no designated sub-region within the TNC 2016 report card, as no previous publicly available dataset existed for it prior to the 2018 surveys, but would most resemble sub-region 31, defined as "less developed fringing reefs, large areas of low relief hard bottom with numerous gorgonians" (*Kramer et al., 2016*).

Nelson Dockyard National Park (NDNP): The NDNP is a combination marine and terrestrial National Park and has a marine boundary of 18.62 km$^2$ (Fig. 1D). According to the TNC 2016 Coral Reef Report Card, it is designated within sub-region 31, which is defined as "less developed fringing reefs, large areas of low relief hard bottom with numerous gorgonians". The NDNP was declared in 1989 under the National Parks Act (1984) and is a known tourism hub for the island, and is home to several major marinas, resorts and boatyards. The marine area of the NDNP is exposed to the Caribbean Sea on the southern side and is bordered by coastal ecosystems (such as mangrove wetlands, rocky shores, beaches), as well as residential communities and above-mentioned

commercial areas on the northern side. The coral reef systems are a combination of fringing and patch reefs, with few areas dominated by boulder reefs. The NDNP was traditionally managed for its historical and cultural value, with almost no focus on the natural history of the area. However, the National Parks Authority has since embarked on an effort to improve the management of the marine and terrestrial ecological aspects of the park (R. Camacho-Thomas & National Parks Authority, 2019, personal observations). To conduct marine ecological assessments within the park, the AGRRA survey methodology was employed to provide baseline marine ecological data to be used in future management of the park. The project also included surveys of seagrass beds and mangrove wetlands, which are ecologically important contributors to coral reef health.

## Survey methodology

To assess the ecological conditions of the reefs within the subject areas, the AGRRA Benthos and Fish protocols (*Lang et al., 2017*) were employed. All surveyors were trained and certified by AGRRA certified trainers, and in some cases (NEMMA and NDNP) included AGRRA surveyors from other islands. Across the three study regions, 26 sites were surveyed using AGRRA protocol between 2017 and 2019. However, for presentation within this article, all sites 5 m or less in depth were removed to stratify the data by depth and allow for clearer comparisons. Presented in this article are: NEMMA (four sites surveyed in 2017), Redonda (four sites surveyed in 2018) and NDNP (13 sites surveyed in 2019) (Table 3).

AGRRA Benthos method: At each survey site, benthic cover was recorded along six transects by identifying flora, fauna, or substrate that lies under the transect line at 10 cm intervals. Each transect is 10 m in length, (giving a total of 100 points per transect) and were deployed haphazardly on the reef. Macroalgal heights (fleshy and calcareous) were measured, to the nearest mm, along two of the six transects at each site. Additional data, such as coral recruits, macro-invertebrates, presence/absence of diseases and trash were also measured during these surveys but are not included in the results of this study.

AGRRA Fish method: Visual counts and size estimates (in 10 cm increments above 5 cm) of the AGRRA fishes (*Lang et al., 2017*) were recorded along 10 belt transects (30 m × 2 m each) located in the same general habitat as the benthos transects. Similar to the benthic transects, these 10 transects were spread across the reef site haphazardly to provide sufficient coverage across the reef at each site. Fish length data were converted to biomass data using L–W relationships sourced from FishBase (fishbase.in). This global information system on fish provides tools, developed using data from up-to-date studies, that can be used to calculate biological and other parameter types of different groups and species of fish. Additional data describing the topographic complexity were also recorded during the surveys but are not included in the results of this study.

All raw benthic and fish data were entered into the AGRRA database, where summary statistics were produced at the site and transect level. The summary data were used to generate the results in this study. The benthic data were separated into groups of benthic promoters and benthic detractors. Benthic promoters are the reef organisms that facilitate
**Table 3 AGRRA site table.**

| Site | Site code | Date | Depth (m) | Latitude | Longitude | Reef zone: habitat | Exposure |
|---|---|---|---|---|---|---|---|
| NEMMA | A03-02 | July 2017 | 7.3 | 17.18114 | −61.75529 | Fore reef: coral field | Exposed windward |
| NEMMA | A04-02 | July 2017 | 10.3 | 17.18958 | −61.78912 | Fore reef: coral field | Exposed windward |
| NEMMA | A05-03 | July 2017 | 7.1 | 17.06484 | −61.66710 | Fore reef: coral field | Exposed windward |
| NEMMA | A09-01 | July 2017 | 6.8 | 17.12034 | −61.70920 | Fore reef: coral field | Exposed windward |
| Redonda | RDAB-01 | July 2018 | 9.1 | 16.93439 | −62.34875 | Fore: boulder reef | Exposed leeward |
| Redonda | RDAB-02 | July 2018 | 8 | 16.94296 | −62.34880 | Fore: boulder reef | Exposed leeward |
| Redonda | RDAB-05 | July 2018 | 9.8 | 16.93876 | −62.34819 | Fore reef: spur and groove | Exposed leeward |
| Redonda | RDAB-07 | July 2018 | 9.3 | 16.94500 | −62.34811 | Fore: boulder reef | Exposed leeward |
| NDNP | ABNPA01 | January 2019 | 6.4 | 17.00348 | −62.83197 | Fore reef: coral field | Exposed leeward |
| NDNP | ABNPA03 | January 2019 | 10.3 | 17.00662 | −61.80928 | Fore reef: coral field | Exposed leeward |
| NDNP | ABNPA04 | January 2019 | 8.5 | 17.00720 | −61.80560 | Fore reef: coral field | Exposed leeward |
| NDNP | ABNPA05 | January 2019 | 10.6 | 17.00738 | −61.79767 | Fore reef: coral field | Exposed leeward |
| NDNP | ABNPA06 | January 2019 | 10.6 | 17.00832 | −61.79350 | Fore reef: coral field | Exposed leeward |
| NDNP | ABNPA08 | January 2019 | 8.7 | 17.00173 | −61.77480 | Fore reef: coral field | Exposed leeward |
| NDNP | ABNPA09 | January 2019 | 9.1 | 17.00155 | −61.76713 | Fore reef: coral field | Exposed leeward |
| NDNP | ABNPA10 | January 2019 | 6.4 | 17.00367 | −61.76375 | Fore reef: coral field | Protected leeward |
| NDNP | ABNPA11 | January 2019 | 8.2 | 17.00077 | −61.76150 | Fore reef: boulder reef | Exposed leeward |
| NDNP | ABNPA12 | January 2019 | 10.3 | 16.99837 | −61.75618 | Fore reef: boulder reef | Exposed leeward |
| NDNP | ABNPA13 | January 2019 | 6.8 | 17.00188 | −61.73758 | Fore reef: boulder reef | Exposed windward |
| NDNP | ABNPA15 | January 2019 | 10.3 | 17.00605 | −61.78013 | Fore reef: coral field | Exposed leeward |
| NDNP | ABNPA16 | January 2019 | 9.4 | 17.00168 | −61.74023 | Fore reef: coral field | Exposed windward |

reef growth and allow coral larvae to settle, and include live corals, crustose coralline algae and sparse turf algae (*Lang & Roth, 2019*). Benthic detractors are benthic organisms like macroalgae, turf algal sediment mats and certain invertebrates (e.g., some sponges, cnidarians, tunicates) that can displace corals or prevent the settlement of coral larvae (*Lang & Roth, 2019*). Fish data were summarized by total fish, commercial species and herbivores (further separated into Scaridae family, Acanthuridae family and other herbivores) biomass. Graphs were plotted to compare results, and where applicable, standard deviation of the means were displayed using error bars. Analysis of Variance (ANOVA) tests were conducted to examine any differences between data averages. Where significant differences were indicated, a Post Hoc Tukey HSD test was used to identify which means varied significantly. All statistical analyses were carried out using KaleidaGraph Statistical Software (Table 4).

# RESULTS

## Benthic results

### North east marine management area

Live coral (LC) percent (%) cover for the NEMMA area ranged from a low of 5% to a high of 21% with an average of 13% while crustose coralline algae (CCA) ranged from

**Table 4 Statistical analysis table for AGRRA surveys.**

| Benthic | | LC | CCA | TAS | MA |
|---|---|---|---|---|---|
| ANOVA | *P*-value | *0.0111* | *0.005* | *<0.0001* | *0.0433* |
| Tukey's HSD | NEMMA vs NDNP | *0.0108* | *0.0066* | *<0.0001* | *0.0344* |
| | NEMMA vs Redonda | 0.4037 | 0.5538 | 0.5232 | 0.2699 |
| | Redonda vs NDNP | 0.2440 | 0.0958 | *<0.0001* | 0.7341 |

| Fish | | TF | CS | HB | Scaridae | Acanthuridae |
|---|---|---|---|---|---|---|
| ANOVA | *P*-value | *0.0262* | *0.0122* | 0.174 | 0.13467 | 0.0938 |
| Tukey's HSD | NEMMA vs NDNP | *0.0205* | *0.0171* | 0.2607 | 0.9915 | 0.0824 |
| | NEMMA vs Redonda | 0.2417 | 0.7015 | 0.9895 | 0.2181 | 0.9856 |
| | Redonda vs NDNP | 0.6367 | 0.1241 | 0.3353 | 0.1344 | 0.2237 |

Note:
Benthic analysis (% cover): LC, live coral; CCA, crustose coralline algae; TAS, turf algal sediment; MA, macroalgae. Fish analysis (g/100 m$^2$): TF, total fish; HB, herbivorous fish; CS, commercial species; Scaridae, scaridae family; Acanthuridae, acanthuridae family. Italicized text represent significant *p*-values ($p \leq 0.05$).

4% to 12% with an average of 9%. LC cover exceeded CCA for all sites (Fig. 2A). Turf algal sediment (TAS) percent (%) cover ranged from 5% to 18% with an average of 13%. Fleshy and calcareous macroalgae (MA) percent (%) cover ranged from 21% to 43% with an average of 31%. MA exceeded TAS for all sites (Fig. 3A).

### Redonda

Live coral percent (%) cover for Redonda ranged from 2% to 17% with an average of 9%. CCA percent (%) cover ranged from 2% to 12% with an average of 7%. LC exceeded CCA for all sites except for Site Code: RDAB-07 (Fig. 2B). TAS percent (%) cover ranged from 0 to 9%, with an average of 3%. MA percent (%) cover ranged from 6% to 31% with an average of 22%. MA exceeded TAS for all sites with the exception of Site Code: RDAB-01 (Fig. 3B).

### Nelson dockyard national park

Live coral percent (%) cover ranged from 3% to 8% with an average of 6%. CCA percent (%) cover ranged from 1% to 9% with an average of 3%. LC exceeded CCA for all sites apart from Site Codes: ABNPA 12 and ABNPA 13 (Fig. 2C). TAS percent (%) cover ranged from 14% to 66% with an average of 52%. MA percent (%) cover ranged from 6% to 30% with an average of 18%. TAS exceeded MA for all sites (Fig. 3C).

## Fish results

### North east marine management area

Total fish (TF) biomass ranged from 2,250 g/100 m$^2$ to 4,595 g/100 m$^2$ with an average of 3,288 g/100 m$^2$. Commercial species (CS) (Appendix 1) biomass averaged 796 g/100 m$^2$ with a low of 333 g/100 m$^2$ to a high of 1,251 g/100 m$^2$ (Fig. 4A). Herbivore (HB) biomass averaged 2,348 g/100 m$^2$ (Scaridae: 1,530 g/100 m$^2$, Acanthuridae: 772 g/100 m$^2$, Fig. 5A), with a high of 3,613 g/100 m$^2$ and a low of 1,240 g/100 m$^2$. HB biomass exceeded CS biomass for all sites apart from Site Code: A05-03 (Fig. 4A).

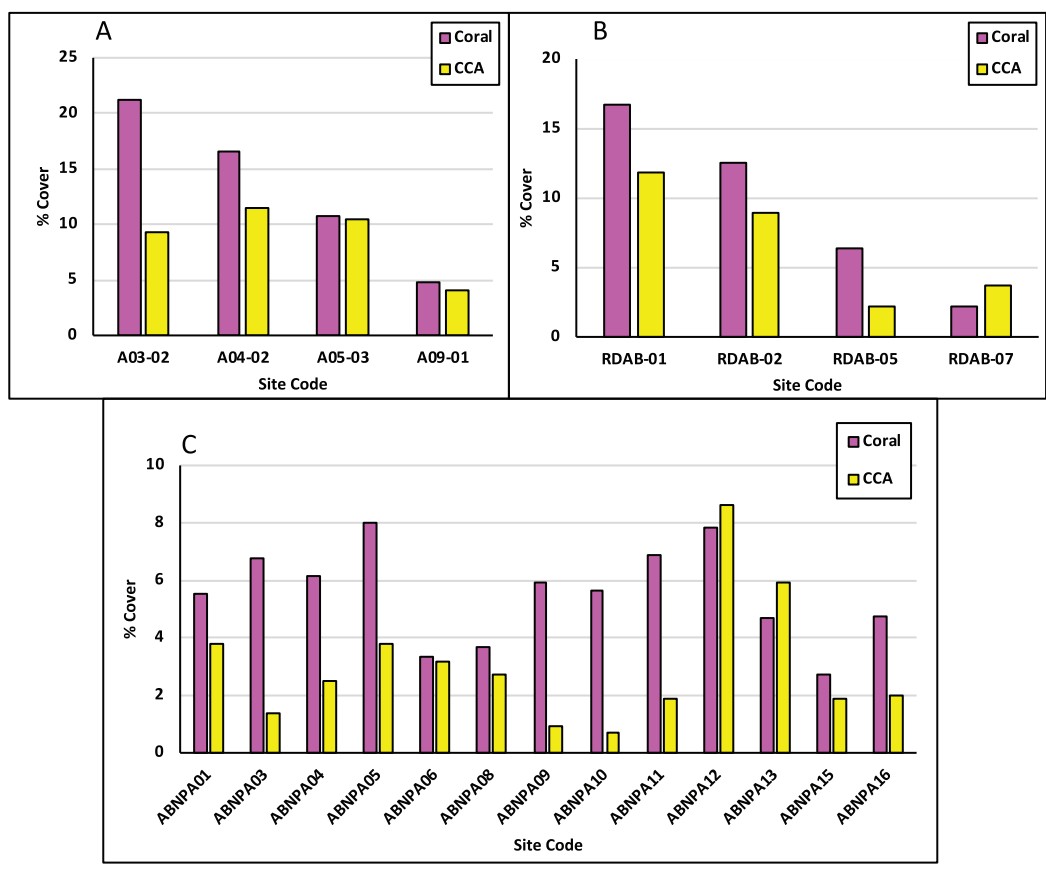

**Figure 2 Benthic promoters across assessment areas.** Each parameter is shown as the average percentage (%) cover in NEMMA (A); Redonda (B) and NDNP (C). LC, live coral; CCA, crustose coralline algae.

### Redonda

Total fish biomass averaged 6,522 g/100 m$^2$ and ranged from 3,659 g/100 m$^2$ to 8,689 g/100 m$^2$. CS biomass averaged 1,621 g/100 m$^2$ and ranged from 645 g/100 m$^2$ to 2,791 g/100 m$^2$ (Fig. 4B). HB biomass averaged 2,467 g/100 m$^2$ (Scaridae: 562 g/100 m$^2$, Acanthuridae: 1,634 g/100 m$^2$, Fig. 5B), ranging from 1,346 g/100 m$^2$ to 3,779 g/100 m$^2$. HB biomass exceeded CS biomass for all sites with the exception of Site Code: RDAB-07 (Fig. 4B).

### Nelson dockyard national park

Total fish biomass averaged 7,953 g/100 m$^2$ and ranged from 2,524 g/100 m$^2$ to 14,909 g/100 m$^2$. CS biomass ranged from 671 g/100 m$^2$ to 6,931 g/100 m$^2$ and averaged 3,193.4 g/100 m$^2$ (Fig. 4C). HB biomass averaged 3,326 g/100 m$^2$ (Scaridae: 1,474 g/100 m$^2$, Acanthuridae: 1,700 g/100 m$^2$, Fig. 5C), and ranged from 1,698 g/100 m$^2$ to 6,171 g/100 m$^2$. HB biomass exceeded CS biomass for six of the 13 sites surveyed (Fig. 4C).

### Overall results

Average live coral percent (%) cover for Antigua, for the surveys carried out in 2017, 2018 and 2019, was 9%, with significant differences between the average coral cover at NEMMA

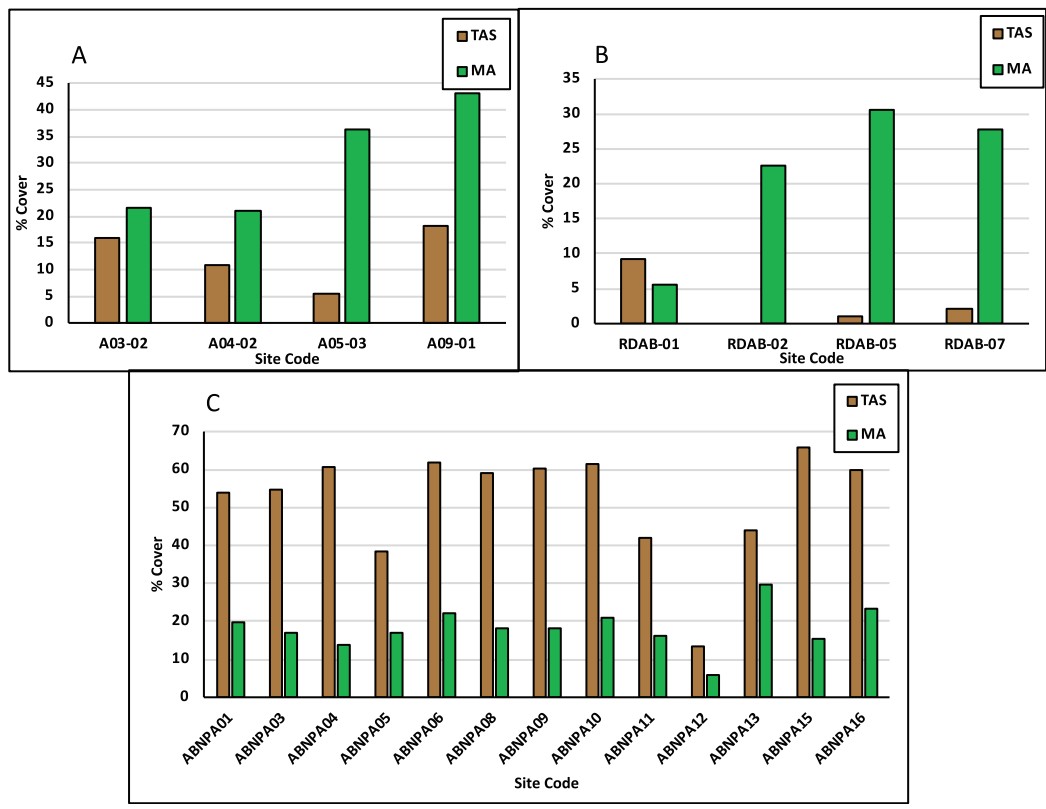

**Figure 3 Benthic detractors across assessment areas.** Each parameter is shown as the average percentage (%) cover in NEMMA (A); Redonda (B) and NDNP (C). TAS, turf algal sediment; MA, macroalgae.

vs NDNP ($p$ = 0.0108). CCA averaged 6%, with significant differences observed between NEMMA and NDNP ($p$ = 0.0066) (Table 3; Fig. 6A). TAS averaged 23%, with significant differences observed between NDNP and Redonda ($p$ < 0.0001), along with NDNP and NEMMA ($p$ < 0.0001). Macroalgal cover averaged 23%, with significant difference seen between NDNP and NEMMA ($p$ = 0.0344) (Table 3; Fig. 6B).

Total fish biomass averaged 5,921 g/100 m$^2$, with significant difference in biomass seen between NDNP and NEMMA ($p$ = 0.0205). Among the commercial species (CS), the average biomass was 1,914 g/100 m$^2$, with significant differences in biomass observed between NDNP and NEMMA ($p$ = 0.0171) (Table 3; Fig. 7A). Herbivorous fish biomass averaged 2,765 g/100 m$^2$, with no significant differences in biomass seen between the assessments. Further analyzed to identify primary herbivores, Scaridae biomass averaged 1,189 g/100 m$^2$ while Acanthuridae biomass averaged 1,369 g/100 m$^2$. No significant difference was observed between Scaridae biomass or Acanthuridae biomass at any of the assessment locations (Table 3; Fig. 7B).

## DISCUSSION

A major issue faced by Small Island Developing States (SIDS) like Antigua and Barbuda is insufficient data availability to provide enough guidance for designation and effective management of Marine Protected Areas. The 2016 TNC Coral Reef Report Cards

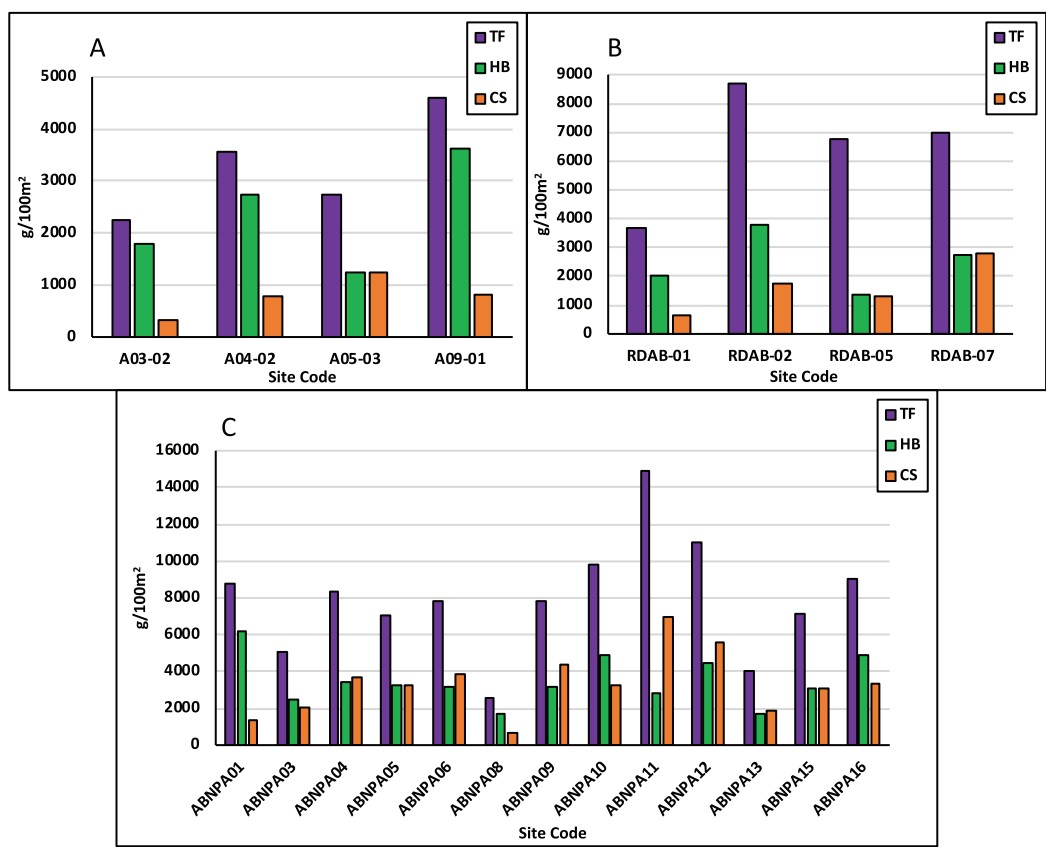

**Figure 4 Fish biomass across assessment areas.** Fish biomass is displayed as grams per 100 m² in NEMMA (A); Redonda (B) and NDNP (C). The following fish groups are displayed: TF, total fish; HB, herbivorous fish; CS, commercial species.

attempted to address this issue by summarizing regional pre-existing datasets for different islands in the Caribbean into unique reports for each island. The goal of the TNC analysis was to create ecological reports of coral reef conditions across islands with similar locations and pressures, which were comparable and easily digestible for decision-makers. However, it was not a targeted effort to provide the resources (financial and technical), which would allow for local stakeholders to assess ecological conditions in current and future MPAs. AGRRA has provided a useful platform for allowing Caribbean SIDS to assess and better understand the ecological conditions of their marine ecosystems. AGRRA provides regional training for personnel for the use of the AGRRA protocol and has an online database, which allows for data comparison not only within a territories waters', but also throughout the region. To aid in the data investigation and comparisons, the AGRRA developers provide assistance in basic data analysis methods and provide data-related GIS products. Currently, several trained AGRRA surveyors exist within the island of Antigua and Barbuda, who are skilled in various protocols. Despite the presence of trained surveyors within the island, ecological surveys are normally dependent on availability of financial (normally through project grants) and human resources, ensuring trained personnel are able to conduct surveys at the indicated time. The surveys conducted

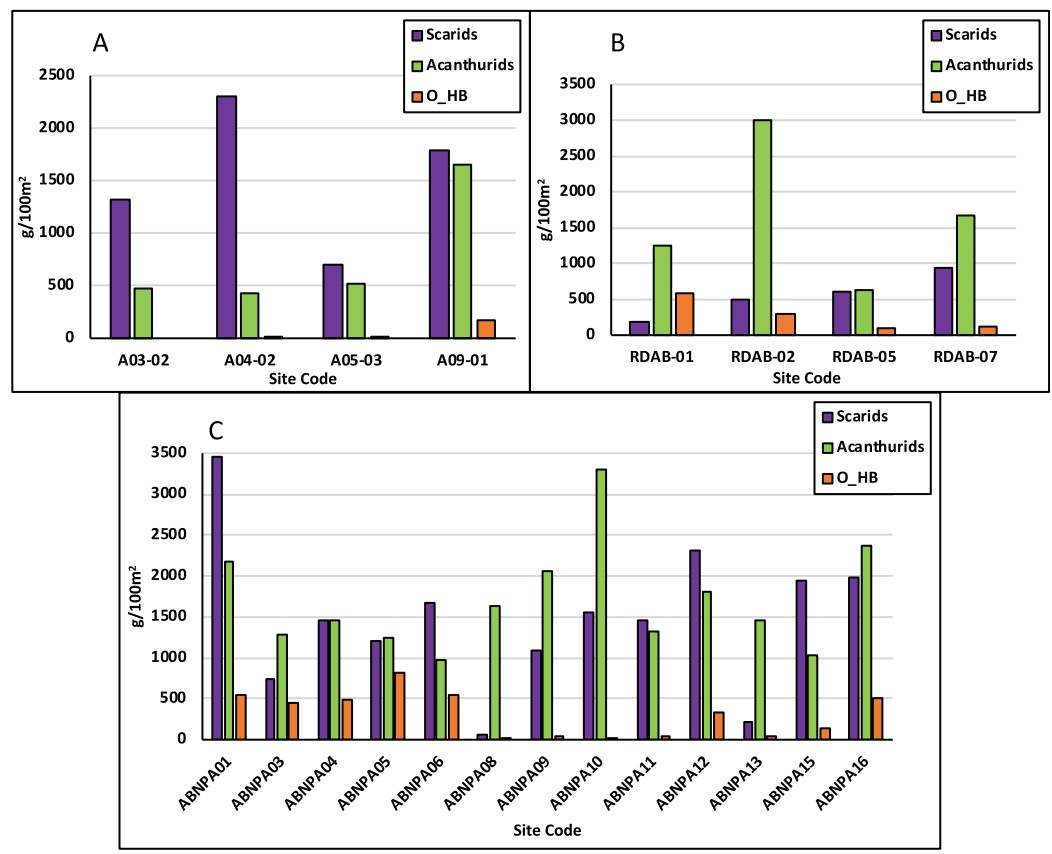

**Figure 5 Herbivorous fish biomass comparissons.** Fish biomass is displayed as grams per 100 m² in NEMMA (A); Redonda (B) and NDNP (C). The following fish groups are displayed: Scarids, scaridae family; Acanthurids, acanthuridae family; O_HB, other herbivores.

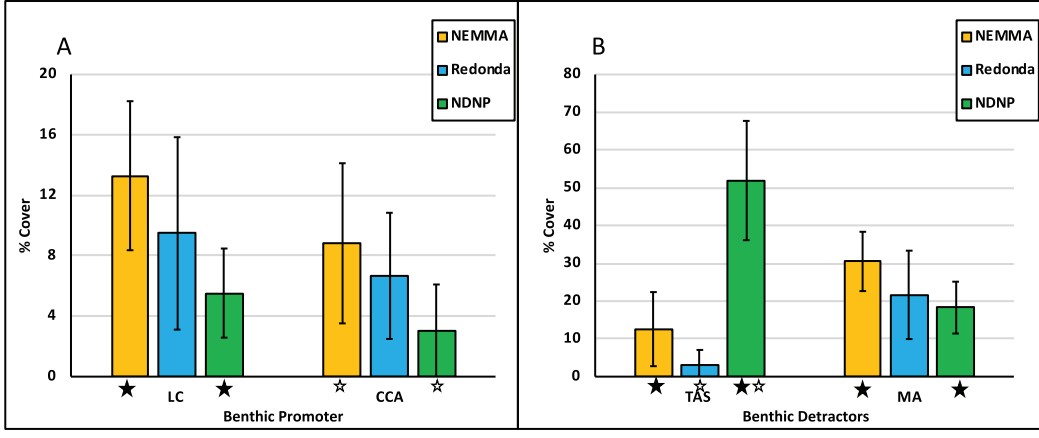

**Figure 6 Benthic promoters vs Benthic detractors.** Benthic parameters are displayed across assessments areas as percentage (%) cover. Benthic promotors (A) are Live coral (LC) and Crustose coralline algae (CCA). Benthic detractors (B) are Turf algal sediment (TAS) and Macroalgae (MA). Error bars are standard deviation of the mean. Significant differences ($p <= 0.05$) are indicated by asterisk (*).

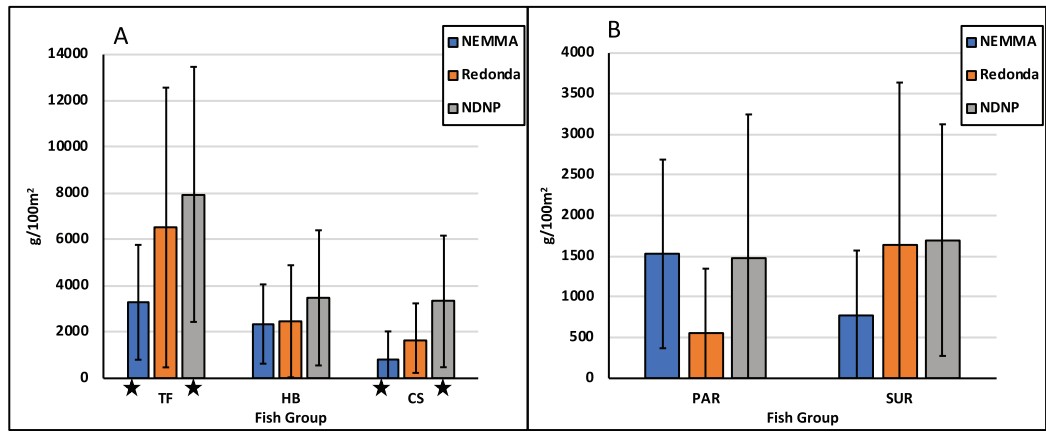

**Figure 7 Fish group comparison.** Fish groups are displayed across assessments areas as biomass (g/100 m$^2$). Fish group (A) are Total fish (TF), herbivorous fish (HB) and Commercial species (CS). Fish group (B) are Scaridae family (PAR) and Acanthuridae family (SUR). Error bars are standard deviation of the mean. Significant differences ($p <= 0.05$) are indicated by asterisk (*).

in the NEMMA, NDNP and Redonda resulted from the expression of need by the local government to inform and/or improve management prescriptions and were funded through various grants.

These surveys and analyses illustrated the high intra-site ecological differences within each assessment, which is highlighted in Fig. 2 (Benthic Promotors), Fig. 3 (Benthic Detractors) and Figs. 4 and 5 (Fish Biomass Comparisons). Sites such as A03-02 had live coral cover recorded at over 20% (Fig. 2A), which was attributed to a proliferation of *Acropora prolifera* stands at this site, something which has been seen in the neighboring island of Guadeloupe (*Japaud, Fauvelot & Bouchon, 2014*). Despite few isolated high coral cover sites, the total average live coral cover was measured at 9% when calculated across all study regions presented in this article (Table 1). A03-02 has been earmarked for further surveys to better understand the factors influencing the proliferation of Acroporids in this site, as well as to investigate its potential future use as a source site for coral restoration in other portions of the island. Crustose coralline algae (CCA), a known positive recruitment influencer for juvenile corals on the reef ecosystem, varied tremendously between assessments, but had an average cover of 6%. Macroalgae was the dominant benthic detractor in NEMMA and Redonda. This however changed in NDNP where the dominant benthic detractor was turf algae (TA) infused with sediment to create a sediment mat (TAS) (Fig. 3). The TAS mat can reduce herbivory by restricting the feeding of parrotfish (Scaridae) species, and affects coral recovery by locking the reefs in an alternative state that does not promote coral growth (*Bellwood & Fulton, 2008*), and could be a contributing factor to the low benthic promotors observed in the NDNP. The extent of this relationship was not explored in this article but has been identified as an area for future studies. Sites with the lowest benthic detractors in the NDNP (Site Code: ABNPA 12) also had the highest benthic promotors, and a similar relationship was seen in several other site results from the NDNP surveys

(ABNPA05, ABNPA11, ABNPA13), as well as the NEMMA (A03-02 and A04-02) and Redonda (RDAB-01 and RDAB-02) surveys (Figs. 2 and 3). The site level information of the benthic promoters and benthic detractors will be utilized in the planning, revision and zoning of these managed areas. Identification of the areas with the greatest reef promotors can help to determine suitable areas to conduct coral restoration experiments and establish conservation zones with the goal of promoting the positive ecological drivers, which should lead to a healthier reef ecosystem.

Further unevenness was also illustrated in the fish biomass comparisons, with total fish (TF) biomass ranging from as low as 2,250 g/100 m$^2$ in the NEMMA region to a high of 14,909 g/100 m$^2$ in NDNP region (Fig. 4). When considering the group dynamics of fish biomass, with the focus on herbivorous (HB) and commercial fish (CS) species (Appendix 1), HB exceeded CS in most sites, with few exceptions in each assessment area, greatest of which was seen at NDNP (Site Code: ABNPA11). Further analysis of the HB biomass illustrated that the Scaridae family was the dominant herbivore group in NEMMA, while the NDNP surveys illustrated mixed variation among all sites. Redonda proved unique as it illustrated a higher proportion of Acanthuridae family to Scaridae family at all sites, due in part to the large schools of surgeonfish observed during the surveys. Concern has been registered, however, regarding the lack of larger bodied Scaridae (vs smaller bodied Acanthuridae) observed in the marine habitat, particularly considering the important role of these species in algal regulation (Lokrantz et al., 2008) (Table 5). Understanding these ecological differences within protected areas can play an important role in ensuring effective management decisions are made for these areas. The data collected from this study can aid reef managers in establishing fish conservation zones and provide evidence for developing fisheries regulations and restrictions within the protected areas. Additionally, all information collected will be utilized as a baseline to understand future studies, establish monitoring protocols, and assist with development decisions.

A high variability between survey results for each area emphasized the differences between assessments. ANOVA analysis (Table 3) showed that there were significant differences between assessments for each category (promotor and detractors) of the benthic characteristics (Fig. 6). Significant differences were also seen in fish biomass of the assessments, when looking at total fish biomass, as well as isolating the commercial species biomass group. However, no significant differences were seen between biomass of herbivore groups (Fig. 7). Using the RHI as a tool to compare ecological assessments, there are some changes between the TNC 2016 Report Card for Antigua and Barbuda and the AGRRA surveys described above (Table 1). Although the TNC and AGRRA datasets cannot be directly compared due to differing data collection methods and sampling design, the ecological characteristics are quantified in the same way providing a useful comparison. On a nation-wide level, coral cover has remained virtually the same, indicating no major loss since the 2016 report cards, which may be attributed to the slow growth rates of the brain corals, which dominate the landscape around the island (R. Camacho, 2017, 2018, 2019, personal observations). However, it also indicates the low impact that bleaching events and coral diseases, such as the Stony Coral Tissue Loss Disease, which have not yet been observed in Antigua and Barbuda (AGRRA, 2019), are

**Table 5 Parrotfish size frequency.**

| Site | Scientific name | Common name | Size batch (cm) | | | | | |
|------|-----------------|-------------|-----|------|-------|-------|-------|-----|
| | | | 0–5 | 6–10 | 11–20 | 21–30 | 31–40 | >40 |
| NEMMA | Scarus iseri | Striped Parrotfish | 80 | 343 | 102 | 3 | 0 | 0 |
| NEMMA | Scarus taeniopterus | Princess Parrotfish | 5 | 68 | 27 | 2 | 0 | 0 |
| NEMMA | Scarus vetula | Queen Parrotfish | 0 | 0 | 11 | 3 | 0 | 0 |
| NEMMA | Sparisoma atomarium | Greenblotch Parrotfish | 1 | 9 | 0 | 0 | 0 | 0 |
| NEMMA | Sparisoma aurofrenatum | Redband Parrotfish | 3 | 19 | 36 | 6 | 0 | 0 |
| NEMMA | Sparisoma chrysopterum | Redtail Parrotfish | 0 | 9 | 14 | 5 | 0 | 0 |
| NEMMA | Sparisoma rubripinne | Yellowtail Parrotfish | 0 | 3 | 5 | 0 | 0 | 0 |
| NEMMA | Sparisoma viride | Stoplight Parrotfish | 38 | 53 | 60 | 38 | 1 | 0 |
| Redonda | Scarus / Sparisoma | Juvenile Parrotfish | 27 | 0 | 0 | 0 | 0 | 0 |
| Redonda | Scarus iseri | Striped Parrotfish | 0 | 0 | 0 | 2 | 0 | 0 |
| Redonda | Scarus taeniopterus | Princess Parrotfish | 0 | 0 | 2 | 0 | 0 | 0 |
| Redonda | Sparisoma aurofrenatum | Redband Parrotfish | 0 | 0 | 5 | 2 | 2 | 0 |
| Redonda | Sparisoma chrysopterum | Redtail Parrotfish | 0 | 0 | 8 | 8 | 0 | 0 |
| Redonda | Sparisoma rubripinne | Yellowtail Parrotfish | 0 | 0 | 4 | 8 | 0 | 0 |
| Redonda | Sparisoma viride | Stoplight Parrotfish | 4 | 1 | 0 | 5 | 0 | 0 |
| NDNP | Scarus guacamaia | Rainbow Parrotfish | 0 | 0 | 1 | 0 | 0 | 0 |
| NDNP | Scarus iseri | Striped Parrotfish | 57 | 12 | 49 | 10 | 0 | 0 |
| NDNP | Scarus taeniopterus | Princess Parrotfish | 0 | 4 | 5 | 14 | 1 | 0 |
| NDNP | Scarus vetula | Queen Parrotfish | 0 | 1 | 4 | 11 | 5 | 0 |
| NDNP | Sparisoma aurofrenatum | Redband Parrotfish | 22 | 39 | 100 | 55 | 1 | 0 |
| NDNP | Sparisoma chrysopterum | Redtail Parrotfish | 0 | 2 | 9 | 47 | 23 | 0 |
| NDNP | Sparisoma rubripinne | Yellowtail Parrotfish | 0 | 1 | 14 | 14 | 16 | 0 |
| NDNP | Sparisoma viride | Stoplight Parrotfish | 3 | 5 | 5 | 29 | 14 | 0 |

currently having on the coral reef ecosystems of the island. Fleshy macroalgal percent cover, on average, was higher than observed in the TNC analysis, which is shadowed by a decrease in Herbivorous fish biomass, despite season limits being placed on parrotfish in 2013 following a noticeable decline in catch numbers in Antigua and Barbuda (*Horsford, 2014*). There have been several studies looking at the relationship between herbivorous fish biomass and fleshy macroalgae coverage (*Mumby & Steneck, 2008*; *Mumby et al., 2012*), and the subsequent negative cascading effect that proliferation of fleshy macroalgae can have on the recruitment of juvenile corals (*Arnold, Steneck & Mumby, 2010*) and the ability of adult corals to grow (*Rasher & Hay, 2010*). Additionally, as *Vallès & Oxenford (2014)* have demonstrated, the analysis of parrotfish body size can be utilized as an indicator of fishing pressure, which will be useful in assessing management effectiveness of these protected areas in the future (Table 5). Commercial species (Appendix 1) biomass, a collation of species with commercial value across the region, displayed a positive trend with an increase in biomass from 2015 to 2019 across surveys (Table 2). One factor contributing to this could be the establishment of closed seasons as nation-wide fisheries management measures implemented by the Fisheries Division (*Fisheries Division, 2013*).

Further management prescriptions; such as limitation of gear types, greater enforcement of closed seasons, or even lengthening of closed seasons; may become essential to enhance the recovery of these species, particularly when considering the importance of key herbivore species (parrotfish) to the coral reef ecosystem.

The information collected during these three reported AGRRA surveys will be directly utilized in the creation of management prescriptions aimed at improving the management of previously established protected areas, and creation of new protected areas. Additionally, these data will serve as an ecological baseline to assess changes and damages to the ecosystem over time. The NEMMA information will be incorporated into the process of updating the management plan for the protected area. Although NEMMA was established in 2005, the management plan was not created until 2007 (*Jackson, 2008*) and did not utilize the data assessed in *Brandt et al. (2005)*. The management plan was based on a series of rapid ecological studies which predominantly provided presence/absence data of species (*Jackson, 2008*). The information collected from our AGRRA surveys will also be used to identify areas of greatest conservation need, such as hotspots (areas of unusually high coral cover) for further research. The NDNP is a national park, which has traditionally been managed from a cultural/historical perspective, with little focus on the marine environment. Recognition of the various threats being faced as a result of a growing economy and an increasingly volatile climate, has led to a greater emphasis on the management of the environmental (marine and terrestrial) resources within the park. The implementation of AGRRA surveys in 2019 represented the first extensive marine data collection at the NDNP. These datasets will be used to inform management of the marine resources using an Ecosystem Based Management (EBM) approach, which incorporates the connectivity of coral reefs and associated ecosystems (*Steneck et al., 2009*). Information about benthic ecological drivers (promotors and detractors) along with fish abundance and other ecological parameters will be used to aid the process of zoning, identification of suitable areas for coral and other ecosystem restoration activities, and improvement of the management of the marine ecosystem.

Redonda's story is a unique one, as the island had been abandoned for many years after previously being utilized as a mining area for guano. Invasive alien species (IAS) such as rats and goats were introduced, and as a result wreaked havoc on the terrestrial fauna and flora. The island has been recognized as an important biodiversity hotspot in the Eastern Caribbean for its importance to nesting seabirds (e.g., Brown Boobies), as well as its endemic species. The Redonda Restoration Program was initiated to remove the IAS and restore the island to its former glory, in a bid to designate it as a protected area (Redonda Ecosystem Reserve). During this process, it was recognized that there is a need to better understand the ecology of the surrounding marine environment. This included not only the nearshore areas surveyed using the AGRRA protocol, but also the deeper coral bank, which surrounds the island, of which some information has been collected using drop-camera surveys (not reported in this article). The rationale here is that the information collected would help to prioritize management activities and zonation of the marine area, which when approved will encompass one of the largest MPAs in the Eastern

Caribbean. Moreover, the marine data provide a useful baseline for future studies of the impact that the islands' terrestrial recovery following the removal of IAS has on the marine ecosystem, particularly considering the results of similar scenarios in the Indian Ocean (*Graham et al., 2018*) and the unique situation created by the low anthropogenic pressure on Redonda.

This article looks at the results of three non-related data collection missions to establish a useful baseline for the island of Antigua and Barbuda. We recognize that the data presented have several limitations, which are primarily driven by the lack of funding and resources to implement a larger marine data collection program. We lack some potentially crucial ecological (such as exposure, depth), methodological (such as time of surveys, number of sites surveyed per assessment) as well as anthropogenic (proximity of commercial institutions, pollution and water quality, fishing pressure, presence/absence of enforcement, etc.) differences, which might help to explain some of the variation seem among and within assessments. Other limitations to the establishment of this dataset as a true baseline for Antigua and Barbuda is the lack of Barbuda data present in the survey. This gap has been identified as an area that needs to be addressed for addition to this growing dataset, and to inform management plans. Additionally, while AGRRA has been identified as the principal coral reef assessment methodology for Antigua and Barbuda going forward, there is still a need for a greater understanding of the role that seagrass beds and mangrove wetlands have on the management of these coral reef ecosystems. By combining these ecological datasets with socio-economics, it is possible to have an EBM approach to the management of these marine areas. What this article represents is an initial effort of the GoAB to not only attempt to better understand the marine ecological conditions affecting the nation's coral reefs through standardized marine data collection, but also a concerted effort to use a holistic approach in the management of marine ecosystem through the incorporation of site level information to inform decision-making.

## CONCLUSIONS

Overall, these surveys indicated that the current status of coral reefs in Antigua and Barbuda are reflective of what was observed throughout the wider Caribbean region, and greater management efforts are needed to improve the overall health of these ecosystems. The high inter- and intra-assessment variability between coral reefs locations surveyed highlights the importance of site level data to guide the management prescriptions for these ecosystems. With increasing pressures from anthropogenic and natural influences, it is important to fully understand the variability between study areas, the impact of stressors and how the management prescriptions will differ appropriately.

Future work will focus on increasing coral reef survey efforts around the nation, with emphasis on those areas within designated or proposed MPAs. Additionally, there are plans to establish permanent monitoring sites within these MPAs to increase understanding of the coral reef ecosystem and its reaction to external pressure and management interventions, with the aim to improve the health of coral reef ecosystems around the island.

## ACKNOWLEDGEMENTS

We would like to acknowledge the following persons, groups and institutions who assisted with data collection: Mr. J. Murphy and Mrs. R. Camacho-Thomas from the National Parks Authority; Mr. T. Joseph and the team at Fisheries Division; Dr. S. Palmer of UWI, Mona; The Redonda Restoration Program Team; and Ms. M. Wilson. Dr. J. Lang and the AGRRA team for their continued support. Mr. J. Williams and Ms. R. Spencer of the Department of Environment for support in the compilation and data analysis of this article. Finally, the Government of Antigua and Barbuda for recognition of the importance in conducting the marine assessments highlighted in this article.

### Funding

The work illustrated here was supported by the Government of Antigua & Barbuda, the Redonda Restoration Program, The Nature Conservancy, CamPAM ToT grant, and the National Park Authority in Antigua & Barbuda. The funders had no role in study design, data collection and analysis, decision to publish, or preparation of the manuscript.

### Grant Disclosures

The following grant information was disclosed by the authors:
Government of Antigua & Barbuda.
The Nature Conservancy, CamPAM ToT.
National Park Authority in Antigua & Barbuda.

### Competing Interests

Ruleo Camacho is employed by the Department of Environment. Sophia Steele is employed by Fauna & Flora International. Shanna Challenger is employed by the Environmental Awareness Group. Mark Archibald is employed by the Fisheries Division.

### Author Contributions

- Ruleo Camacho conceived and designed the experiments, performed the experiments, analyzed the data, prepared figures and/or tables, authored or reviewed drafts of the paper, and approved the final draft.
- Sophia Steele conceived and designed the experiments, performed the experiments, authored or reviewed drafts of the paper, and approved the final draft.
- Shanna Challenger conceived and designed the experiments, performed the experiments, authored or reviewed drafts of the paper, and approved the final draft.
- Mark Archibald conceived and designed the experiments, performed the experiments, authored or reviewed drafts of the paper, and approved the final draft.

### Data Availability

Raw data (fish data and Benthic data) are available in the Supplemental Files.

## Supplemental Information

Supplemental information for this article can be found online at http://dx.doi.org/10.7717/peerj.9236#supplemental-information.

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
