# Peer review of "Status of coral reefs in Antigua & Barbuda: using data to inform management"

_PeerJ, doi:10.7717/peerj.9236_

## Round 0.1 · original submission · Major Revisions

Three expert reviewers have evaluated your manuscript and the comments can be seen below. As you can see, the reviewers are positive about the manuscript, but all have pertinent and insightful observations that will help to improve the manuscript.

·

Basic reporting

Intro and Background – If the paper is going to reference the 2016 TNC report card for comparison with the AGRRA survey results, more information is needed about the surveys used for the report card.

Figures – A map of all MPAs and sites should be included. I would recommend including the sites used to develop the TNC report card in the map as well. Figure 3 should be split into two tables in order to avoid misinterpreting the columns.

Experimental design

This paper compares data across NDNP, Redonda, and NEMMA where the sample sizes are unequal, and the range of habitat types represented in the data is unknown. There needs to be an explanation of how and why the site were chosen within each of these regions. Can four sites accurately represent baseline data for an area as large as the proposed Redonda MPA? If the representation of habitats determined by reef structure, depth, and distance from shore is not considered, the results may not be representative of the true population of the functional groups recorded. Therefore, comparing across the MPA’s will provide skewed results.

If large scale reef habitat types are known such as barrier reefs, patch reefs, and fringing reefs, why were they not included as a variable in these results? Much of the variability identified between sites within each region may be been a result of the varying habitat types.

As stated in the Basic Reporting section above, if results are going to be compared to those reported by TNC, there must be an explanation of how the spatial coverage and sample size of each survey effort differed and what makes them comparable.
If the spatial coverage of the three AGRRA surveys is only representative of a portion of the reefs surrounding Antigua and Barbuda, I would avoid extrapolating the information to make statements about the reef nation-wide.

The AGRRA benthos methods section needs better clarification and needs to identify which data were used in the analysis of the paper. Furthermore, the analysis methods need to include how the data was compiled for each site to generate percentage values.
The methods section would benefit from some additional explanations of the data collected where the terms promotors and detractors are defined.

Validity of the findings

The paper should include a full disclosure of the sampling design and power analysis used to develop sample sizes within each region (MPA).

When sample sizes, habitat representation, and seasonality vary across datasets, I would advise using caution when comparing between them. In addition, due to the variability among the datasets, I would also advise against making inferences or speculations about what might have caused those differences within the discussion of the paper.

Reviewer 2 ·

Basic reporting

The writing itself is OK but there were mistakes. The first several figures are in fact tables. In manuscript literature citations are unconventional but not a serious problem.

Experimental design

Problems are described in general comments to authors

Validity of the findings

Problems of the survey design are described in the general comments to the authors.

Additional comments

General Comments

This is a study of three coral reef systems on or near the island of Antigua (in the country of Antigua and Barbuda). The authors used several established indicators of reef condition such as live coral and crustose coralline algae (CCA) as indicators of reefs in good condition (called reef "promotors") and macroalgae and thick sediment trapping algal turfs as indicators of reefs in poor condition (called reef "detractors") .

The results of this study showed considerable variability within sites but significant differences among regions. For example, the NEMMA region had abundant coral and CCA (i.e., "promotors") but it also had abundant macroalgae detractors. This is just one of several examples of the variable results from this study. The authors clearly acknowledge these confusing patterns.

My primary concerns relate to the lack of stratification in the design of the study and the lack of a clear idea of how managers can reduce "detractors" and increase "promotors" of healthy reefs. I'll describe what I see as problems below but I want to stress up front that Antigua is an important island that had extensive coral reefs but most (all?) have declined over the past several decades. So, studying the state of the island's management is extremely important. However, I found this study frustratingly incomplete. All problems can and should be fixed.

It would have been helpful to see detailed charts showing the location of each of the study reefs within the study regions. Readers are left wondering were the reef surveys conducted on exposed or protected reefs? Were fore reefs, back reefs or patch reefs studied? None of those important variables were described in the manuscript.

Per region water depths were reported but they ranged remarkably from 1.5 m to 10.6 m. Pooling data over such a great range in water depths is a problem especially since the average water depths among regions varied (i.e. NEMMA averaged 5.3 m, NDNP averaged 8.6 and Redonda averaged 9.0. Most rate processes such as algal productivity and herbivory vary considerably over depth ranges that broad. Importantly, both "benthic promoters" of coral cover and CCA abundance are depth sensitive. Tropical CCA are usually most abundant in shallow water (Adey and Vassar 1975, Dean et al 2015 ). Fig. 8A shows the greatest CCA abundance at the shallowest (NEMMA) site. Further, the relatively high coral cover at the NEMMA region was said to be the result of abundant Acropora prolifera which is a shallow water branching coral. This leaves open the question of whether any differences in management or simply differences in water depths resulted in the regional differences illustrated in the table in Fig. 3.

Several important points were not clearly described. For example, the North-East Marine Management Area (NEMMA) is said to be the largest managed marine area on the island of Antigua but there is no indication of what is "managed" and if enforcement and/or compliance exists in this area. Specifically, is this a no-take reserve? Or are there some other restrictions on fishing or other activities? These questions seem central to this paper.

I would have liked to see more data analyzed. What coral species were present? As mentioned above, it was revealed that one site included the branching coral Acropora prolifera. That is important and interesting for two reasons. First that is the genus that built most coral reefs in the Caribbean and that species is a hybrid of two other Acropora species that would have to be on the island but were not mentioned. Other coral species are susceptible to bleaching or disease. It would be good to document (perhaps in an appendix) all coral taxa.

While relatively few studies have been conducted on Antigua's coral reefs, beyond the AGRRA and TNC studies the recently published study in Science Advances (Steneck et al 2018) included several study sites in Antigua. Those might provide additional context for this paper.

Finally, the composition style of the paper could be improved. The first three figures are not figures at all but are tables. Figure 4 is the first true figure. Several words or phrases that are not proper nouns were capitalized. "Macroalgae" was not written as a single word. There were misspelled words such as "assess" (not spelled that way), references were cited in non-traditional ways.


Literature cited:

Adey, W.H. and Vassar, J.M., 1975. Colonization, succession and growth rates of tropical crustose coralline algae (Rhodophyta, Cryptonemiales). Phycologia, 14(2), pp.55-69.

Dean, A.J., Steneck, R.S., Tager, D. and Pandolfi, J.M., 2015. Distribution, abundance and diversity of crustose coralline algae on the Great Barrier Reef. Coral Reefs, 34(2), pp.581-594.


Specific Comments:

English composition and style: Cite only author's last names. The exception is if several authors have the same last name - in which case you include the authors first initial.

Only capitalize proper nouns (e.g., Line 81: "invasive alien species" should be all lowercase. Line 146, 147 " crustose coralline algae" and "coral cover" etc. are common names of organisms and thus are not proper nouns. It is OK to capitalize acronyms.

Abstract becomes confusing towards the end when this study is compared to other studies. It would be clearer if you state "our study" or "we". For example, rather than "While the results of the surveys mirror..." It would be better to write: "While the results of our surveys mirror"..

Line 48: MPA don't generally 'stem the decline' as much as they improve potential recovery after mortality.

Line 60: Figure 1 is a Table. In fact, Figs 1 - 3 are all Tables and should be presented as such.

Line 75: A map that includes all three study sites would be helpful for readers who do not know the area.

Line 128 (and throughout): You used "quadrats" not "quadrants".

Results:

Starting with line 145: Comparing sites having different depths creates a problem. Note that the data for all three study sites are different depths (see the Table mislabeled as Fig. 2). NEMMA average depth is 5.4 m (+ 1.06 SE), Redonda sites average 9.05 m (+ 0.38 SE) and NDNP is 8.6 m (+ 0.5 SE). At the very least for NDNP, the 10 sites > 8m depth could be used to compare with Redonda.

Line 218: The abundance of Acropora prolifera at one site illustrates why this study should be depth stratified. Acropora species grow in greatest abundance in shallow zones where they occupy considerable substrate. The authors describe this at one of the shallower site. However, there is no support in the literature to suggest that outplanting this coral will restore reefs at the scale of the three study reefs in Antigua.

Line 226: The word "macroalgae" is one word not two as in line 226 and not hyphenated e.g. in Fig. 8.

Line 239: The discussion of reef fishes is interesting. However, I didn't see anything describing the method of fishing on these reefs. Where fish traps and spear fishing is most common, herbivorous fishes (especially parrotfishes) decline in abundance. Where hook and line fishing predominates, carnivorous fishes decline.

Line 246: Good point about the increased importance of large bodied parrotfishes. Why wasn't this illustrated to see if that was a factor in this study.

Line 248: High intersite variability is a problem for this study to make clear conclusions. For that reason, stronger stratification of sites is warranted. The remarkable depth range at the NEMMA sites (i.e. going from 1.5 to 10.6 meters) should have been controlled. Perhaps use depth bins. Also, if the sites varied by wave exposure, this would likely affect growth rates of algae and/or grazing rates by herbivores.

Line 257: Why weren't live coral cover per species report? The concern about diseases and bleaching can be better evaluated when species are reported.

Line 281: Marine surveys around Redonda "have been fed directly onto the rationale for the creation of the Redonda Ecosystem Reserve management plan." EXACTLY WHAT WAS USED FOR THAT RATIONALE? That is the sort of information that should be evident in this paper but is not.

·

Basic reporting

I did have a great interest in the research topic addressed in the ms and did enjoy reviewing the material. The manuscript would benefit from some general reorganization and recrafting of the hypotheses before publication. I've provided several general recommendations for the authors to consider that would improve the overall layout and flow of the manuscript while also providing some specific, directed comments that should be included in a revised ms.

Musts in a revised ms.
1. Figures - A map of the study area and study sites. It's important to know the distance between some of the research areas, spatial distribution, and how the location of the sites may affect the results. It would also be beneficial to identify the type of habitat surveyed for each survey site (e.g. patch reef, fringing, etc.)

2. Methods - Statistical interpretation of the data. Several AGRRA techniques are mentioned in the methods (line intercept, quadrats, juvenile coral counts) yet only live coral cover is presented in the results and figures. First, was the only the line intercept data used? Second, if so how was live coral cover calculated and summarized? Were the points pooled for all transects or averaged across transects at each site? How was the data treated for the management type analyses (pooled or averaged?). Third, if data from some of these surveys were not used in the analyses then it can be omitted from the methods. For example, I did not see any data presented on large and small coral juveniles (the authors call these recruits in the methods).

3. Introduction - I understand that this manuscript is presenting baseline data. That is perfectly acceptable and provides a starting point for the study region. The authors continually emphasize this point throughout the ms however they would be better served defining and interpreting the results in the context of the 3 management areas assessed. From the descriptions offered in the Intro and Methods it seems that there is a wide disparity in the management actions between the 3 study areas (NEMMA, Redonda, & NDNP) yet the interpretation of the results and discussion never allude to how the differences in protection, access, or geography might correspond with the differences in coral cover, TAS, or MA cover. Certainly some hypotheses could be generated that would state that live coral cover should correspond with increased protection or more restricted access. These would be a priori assumptions based upon the literature. Regardless of the paucity of long-term data for the study region the present day assessments provided by the authors could still be the result of the current management zone practices that are in place. It doesn't have to be a temporal analysis. The same point can also be made for the results of the fish surveys and integrated together does the highest herbivore fish biomass correlate with the highest coral cover. These are all hypothesis that could addressed in a revised ms.

Experimental design

General considerations

As mentioned above the authors reiterate that this is a baseline study and that there is no long-term data from the study region. However, if they wanted to analyze the decline of corals across the 3 study areas AGRAA demographic surveys do estimate the amount of old mortality as part of the survey. While the metric of live coral tells readers what is currently there the authors could summarize the frequency and the amount of "old mortality" on the corals surveyed to indicate what might have been lost in the 40 years of degradation in the study region.

Figures - The figures should be re-orientated. It's not visually appealing to see the data typically placed on the Y axis to be on the X axis (e.g. fish biomass, coral cover).

Validity of the findings

With the exception of the mention A. prolifera at one of the sites there is little mention of the coral composition either at the management zone level or site level. Are any of the differences across the 3 study areas tested due to coral community composition? Or the habitat (reef types) assessed in each study region? Differences across the 3 study zones might be explained by the presence, absence, or abundance of certain coral species. Density or abundance data would help assist in understanding the difference across the 3 study zones (for example numerous small colonies vs. few large framework building colonies).

---

## Round 0.2 · Minor Revisions

Two reviewers have evaluated your resubmission and both have noted major improvements. However, there are issues with the writing style that should be corrected.

·

Basic reporting

Overall, the context of the paper has greatly improved by providing greater detail in the methods and clarifying differences between the TNC and AGRRA datasets. However, despite the improved context, I feel that the writing technique needs a great deal of editing. Throughout the manuscript, there were an abundance of run-on sentences and back to back compound sentences with differing subjects. In addition, I felt the discussion contained a lot of redundancy in stating what the data was useful for in regard to management actions. There might be some value in making your findings more concise.

A few other grammatical errors I found included improperly capitalizing words (coral reefs, seagrass beds, mangroves, etc.) and switching between past and present tenses.

Once the grammatical errors and writing format have been revisited, I feel that an additional review is needed to finalize the manuscript. Again, I feel that the manuscript has greatly improved from the first version I reviewed and provides informative and important results that will ultimately evolve into a great publication.

Experimental design

My concerns with the experimental design were addressed in my prior review of the manuscript.

Validity of the findings

The manuscript does a great job at identifying the need for basic reef assessments and regular monitoring to establish baseline data and evaluate the status of marine resources to improve management decisions. However, I would suggest adding in some additional detail to the future research recommendations included in the discussion. Stated several times within the manuscript is the need for additional surveys and funding to support those efforts to best understand how to manage a reef system. I feel that researchers may look to this publication for information on where to fill data gaps and what management applications these missing data may support. This is certainly just a suggestion, but I feel that insightful recommendations may generate more interest towards these protected areas of Antiqua and Barbuda.

Reviewer 2 ·

Basic reporting

Good and improved

Experimental design

Major problem with stratification etc has been resolved.

Validity of the findings

Findings are valid

Additional comments

Good job with the revision

---

## Round 0.3 · Minor Revisions

Improvements have been made to the mansucript. However, in reading over the manuscript I encountered a number of errors that need to be attended to. Please see attached document for details. Numbers refer to line numbers in the track changes version of the manuscript.

---

## Round 0.4 · accepted · Accept

I am satisfied with the changes made to the manuscript.